# Dyadic Mamba: Long-term Dyadic Human Motion Synthesis

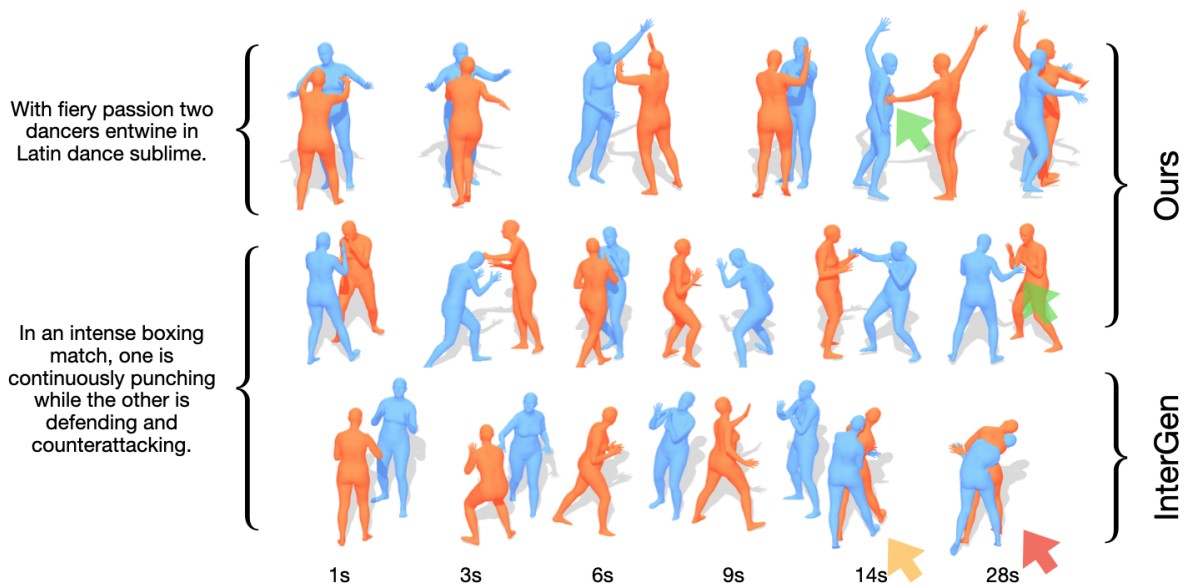

Figure 1. Given a textual description of a dyadic interaction, our model produces the dyadic human motion for the two persons. Our Mamba-based approach is capable of producing longer sequences than state-of-the-art Transformer-based approaches. The first two rows depict generations from our model, while the last row shows a generation from InterGen [18]. Our model successfully generates motion sequences exceeding the training length of 10 seconds. Notably, it maintains close contact between the two individuals over extended time horizons. In contrast, transformer-based approaches like InterGen [18] (last row) struggle with temporal extrapolation, resulting in motion artifacts or even complete breakdown (easier to observe in supplementary video).

## Abstract

*Generating realistic dyadic human motion from text descriptions presents significant challenges, particularly for extended interactions that exceed typical training sequence lengths. While recent transformer-based approaches have shown promising results for short-term dyadic motion synthesis, they struggle with longer sequences due to inherent limitations in positional encoding schemes. In this paper, we introduce Dyadic Mamba, a novel approach that leverages State-Space Models (SSMs) to generate high-quality dyadic human motion of arbitrary length. Our method employs a simple yet effective architecture that facilitates information flow between individual motion sequences through concatenation, eliminating the need for complex cross-attention mechanisms. We demonstrate that Dyadic Mamba achieves competitive performance on standard short-term benchmarks while significantly outperforming transformer-based approaches on longer sequences. Additionally, we propose a new benchmark for evaluating long-term motion synthesis quality, providing a standardized framework for future research. Our results demonstrate that SSM-based architectures offer a promising direction for addressing the challenging task of long-term dyadic human motion synthesis from text descriptions.*

## 1. Introduction

Synthesizing human motion from text is highly relevant for the entertainment industry, such as video games and films, as it can replace costly and time-consuming motion

capture or provide a reasonable starting point for digital artists. Additionally, these models can serve as placeholders before complex and expensive motion captures commence, still conveying some level of artistic intent. One of the most critical and challenging aspects of human motion synthesis is animating social interactions, which require accurately modeling the simultaneous movements of two persons.

Recent progress in the social human motion generation domain has been facilitated by two new text-to-dyadic human motion datasets, InterHuman [18] and Inter-X [44], enabling data-driven approaches to dyadic human motion generation. InterGen [18], a recent transformer-based [42] denoising diffusion model [12], addresses this task using layer-wise cross-attention between the two interacting human motion signals. InterMask [13] is a discrete state-space model which leverages masked motion modeling [4] which obtains impressive results. However, similar to InterGen, InterMask relies on transformers and cross-attention as a model backbone. While transformers and cross-attention are powerful tools for time series synthesis, they struggle with producing long sequences due to limitations in positional encoding and computational complexity. For example, both InterGen and InterMask utilize Absolute Positional Embeddings and train on sequences with at most 10 seconds of length. Social interactions, however, often last for tens of seconds [16] or even minutes, necessitating models capable of generating dyadic motion of arbitrary length. Some recent single-person motion generation works, such as FlowMDM [1], address this with a rolling window approach, resetting positional encodings at fixed intervals. However, this can result in inconsistent motion styles across different windows, which is particularly problematic for dyadic interactions that require careful consistency not only within each person but also between both persons.

Problems with the Absolute Positional Embeddings for sequence lengths extrapolation can be remedied somewhat with Rotary Positional Embeddings (RoPE) [34]. To evaluate this, we extend InterGen by replacing the Absolute Positional Embedding with RoPE. However, in our experiments we verify that InterGen with RoPE cannot extrapolate beyond $2\times$ the training sequence length, which is in line with recent findings on RoPE [5, 15, 23, 28].

To enable us to truly generate motion sequences of arbitrary length, we build on recent advances in State-Space Models (SSMs), specifically Mamba [8], which effectively handle very long sequences. Surprisingly, a relatively simple approach enables us to generate highly competitive results of arbitrary sequence length. As our goal is long-term motion synthesis, we optimize directly in data space, similar to InterGen and in contrast to InterMask. As Mamba has no equivalent of cross-attention, we enable information flow between the individual persons via simple concatena-

tion. Our experiments confirm that our simple yet effective model design outperforms other data-space based methods, while being more parameter efficient.

In Figure 1, we qualitatively compare dyadic motions generated by our Mamba-based approach with a recent transformer-based method. While the transformer-based approach produced artifacts when generating motion beyond its training horizon, our model is capable of synthesizing realistic motion well beyond the training sequence length. To further verify that our model effectively models long-term motion synthesis, we introduce a new easy-to-replicate long-term motion synthesis benchmark based on individual motion quality, which can act as a first step towards long-term motion synthesis.

In summary, our contributions are two-fold: (1) We introduce Dyadic Mamba, a simple yet effective dyadic motion synthesis model, which produces highly competitive results on two dyadic motion synthesis datasets on short-term dyadic motion benchmarks. (2) We introduce a first long-term motion quality benchmark and verify that our method outperforms transformer-based approaches in motion synthesis over long time horizons.

## 2. Related Work

**Single-Person Human Motion Synthesis**: Early works in text-to-motion synthesis leverage GANs [17] and Transformer-based conditional VAEs [10, 25] to map textual descriptions to human motion. MotionCLIP [38] utilizes the pre-trained CLIP [29] space to align human motion with textual descriptions. The introduction of discrete latent spaces [7, 9, 26, 27, 49, 50] spanned by vector-quantized variational auto-encoders (VQ-VAEs) [41] has further enhanced the effectiveness of human motion generators. Generative-Pretrained Transformer (GPT)-based approaches [14, 45] employ GPT-style next-token prediction for motion synthesis while mask prediction models [4, 9, 26] predict randomly masked tokens. Recently, diffusion-based generative models have emerged, with MDM [40] and MotionDiffuse [46] utilizing transformer-based architectures for the text-to-motion task. CondMDI [3] extends these methods for effective motion in-betweening, while MLD [2] leverages a pre-trained latent space to significantly reduce computational costs. ReMoDiffuse [47] integrates a database retrieval system to refine motion generation, and FlowMDM [1] employs blended positional encodings to overcome the limitations of transformers in synthesizing very long motion sequences. Finally, MotionMamba [48] replaces the computationally expensive Transformer with the more lightweight Mamba [8] model, enabling faster motion synthesis.

**Multiple Human Motion Synthesis**: Generating multiple persons is inherently more challenging than generating a single human motion sequence. In the single-person case,

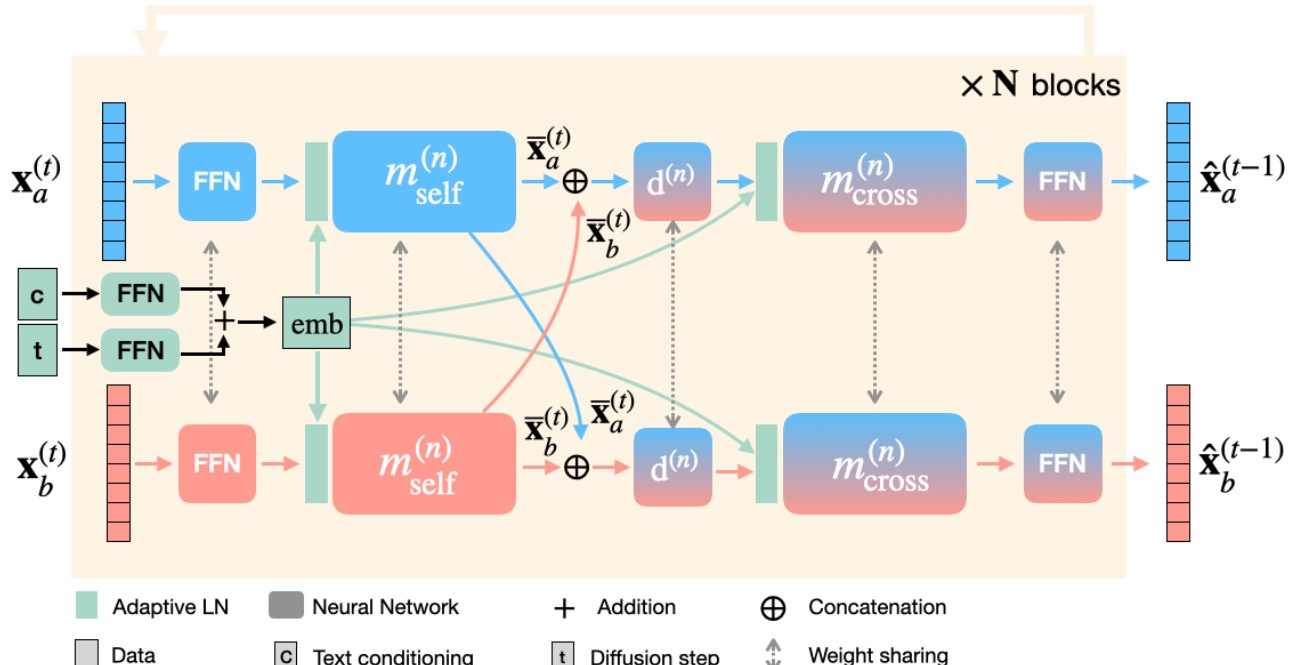

Figure 2. **Model Overview:** The Dyadic Mamba performs single-step denoising of a $t$-noised dyadic signal $\{\mathbf{x}_a^{(t)}, \mathbf{x}_b^{(t)}\}$ to produce the denoised dyadic signal $\{\hat{\mathbf{x}}_a^{(t-1)}, \hat{\mathbf{x}}_b^{(t-1)}\}$. Our architecture comprises $N$ cooperative blocks that process signals iteratively, culminating in the final motion prediction. Each block integrates text embedding $\mathbf{c}$ and diffusion step embedding $t$ through projection and summation, yielding a conditioning embedding emb that modulates the network via Adaptive LayerNorm [22,24]. The processing pipeline begins with individual motion sequences undergoing linear projection to align their dimensionality before being processed by an *individual* Mamba module $m_{\text{self}}^{(n)}$ to obtain $\overline{\mathbf{x}}_\circ^{(t)}$ where $\circ \in \{a, b\}$. Subsequently, the two individual motion sequences $\overline{\mathbf{x}}_a^{(t)}$ and $\overline{\mathbf{x}}_b^{(t)}$ are concatenated and linearly projected by $d^{(n)}$ to re-align the dimensionality. The resulting signal contains information about both persons and is passed to the *cooperative* Mamba block $m_{\text{cross}}^{(n)}$. The final output is obtained through linear projection to the target dimensionality. Notably, parameter sharing across individual motion sequence processing ensures both computational efficiency and order invariance in the dyadic motion representation.

the global transformation can be canonicalized. However, for social interactions, this is infeasible as both individuals must be generated within a shared space. Some motion forecasting approaches [43] address this by predicting in global coordinates, while others employ per-person normalized motion [37]. The former approach is effective only for very short time horizons, whereas the latter is suitable for limited interactions. PriorMDM [32] utilizes a fixed off-the-shelf single-person motion generator coupled with a learnable slim communication block to generate dyadic motion. RIG [35] introduces role-aware interaction generation by assigning roles prior to motion synthesis. FreeMotion [6] introduces a framework that can produce any number of persons utilizing a transformer-based separate interaction and motion-generation module.

Recently, two text-to-dyadic human motion datasets have been released, advancing the field: InterHuman [18], comprising 6.56 hours of dyadic interactions with human poses represented as SMPL [19], and Inter-X [44], which includes extensive dyadic interactions with SMPL-

X [19,21], incorporating finger motions. InterGen [18], proposed alongside the InterHuman dataset, leverages a multi-layered interactive Transformer, where information is exchanged via cross-attention at each layer. InterMask [13] is a discrete state-space model which leverages masked motion modeling [4], which obtains impressive results. In contrast to data-space based models such as InterGen, it requires a two-stage training strategy where first a discrete latent motion representation is learned via a VQ-VAE, which is subsequently used for masked motion modeling. In2In [30] builds on the Transformer-based architecture of MDM and additionally utilizes an LLM to disentangle the individual motion descriptions in the provided text. While transformer-based approaches yield high-quality results, they struggle with scalability as the number of persons or frames increases due to the quadratic complexity of the attention mechanism. Our method, in contrast, relies on Mamba [8], which mitigates this limitation.

## 3. Method

In this work, we model dyadic interactions between two humans as:

$$\{\mathbf{x}_a, \mathbf{x}_b\} = \text{gen}(\mathbf{c}, L), \tag{1}$$

where $\mathbf{c}$ is a conditioning signal, specifically an embedding of a textual description, and $L$ is the desired length of the motion sequences. Human motion sequences of length $L$ are represented as $\mathbf{x}_a = \{\mathbf{x}_a(i)\}_{i=1}^{L}$ and $\mathbf{x}_b = \{\mathbf{x}_b(i)\}_{i=1}^{L}$, where $\mathbf{x}(i) \in \mathbb{R}^{d_{\text{pose}}}$ denotes a human pose representation with dimension $d_{\text{pose}}$ and where $a$ and $b$ indicate the two persons interacting. Our method is agnostic to the pose representation and we show in our experiments that our model performs competitively both on 3D joint representations [18] as well as on articulated poses [44] such as SMPL [21]. We utilize a frozen CLIP [29] text encoder to obtain a text embedding $\mathbf{c} \in \mathbb{R}^{d_{\text{CLIP}}}$, following the approach of InterGen [18]. The generation function $\text{gen}(\mathbf{c}, L)$ is represented as a denoising diffusion model [12], where a dyadic interaction is sampled via reverse diffusion sampling:

$$\mathbf{x}_a^{(T)} \sim \mathcal{N}(\mathbf{0}, \mathbf{I}), \quad \mathbf{x}_a^{(T)} \in \mathbb{R}^{L \times d_{\text{pose}}}$$

$$\mathbf{x}_b^{(T)} \sim \mathcal{N}(\mathbf{0}, \mathbf{I}), \quad \mathbf{x}_b^{(T)} \in \mathbb{R}^{L \times d_{\text{pose}}}$$

$$\{\mathbf{x}_a, \mathbf{x}_b\} = \left\{ \text{denoising}(\{\mathbf{x}_a^{(t)}, \mathbf{x}_b^{(t)}\} | t, \mathbf{c}) \right\}_{t=T}^{1}, \tag{2}$$

where $t$ represents the diffusion step with maximal diffusion step $T$.

### 3.1. Dyadic Mamba

Dyadic Mamba implements the denoising function, $\text{denoising}(\{\mathbf{x}_a^t, \mathbf{x}_b^t\} | t, \mathbf{c})$, necessary for reverse diffusion sampling, to generate a dyadic motion $\{\mathbf{x}_a, \mathbf{x}_b\}$ conditioned on text input $\mathbf{c}$. As illustrated in Figure 2, Dyadic Mamba consists of $N$ stacks of *cooperative blocks*, where each block takes as input a hidden representation with dimension $h$ for each person $a$ and $b$, as well as conditioning signals $t$ and $\mathbf{c}$, and outputs the processed motion for each person. After the last block, the sequences are projected to pose dimension $d^{\text{pose}}$.

**Mamba Module:** Each cooperative block comprises two distinct Mamba modules: $m_{\text{self}}^{(n)}$ focusing on individual motion dynamics and $m_{\text{cross}}^{(n)}$ addressing cooperative motion between persons $a$ and $b$. Here, $(n)$ represents the block index. For clarity, we omit the block indicator $n$ and the feed-forward networks (FFN) that adjust data dimensionality in subsequent descriptions.

For dyadic motion processing, we first compute intermediate motion representations:

$$\bar{\mathbf{x}}_a^{(t)} = m_{\text{self}}\big(\text{AdaLN}(\mathbf{x}_a^{(t)} | \mathbf{c}, t)\big) \tag{3}$$

$$\bar{\mathbf{x}}_b^{(t)} = m_{\text{self}}\big(\text{AdaLN}(\mathbf{x}_b^{(t)} | \mathbf{c}, t)\big). \tag{4}$$

These intermediate representations capture individual motion characteristics without cross-person information exchange. The functions $m_{\text{self}}$ and $\text{AdaLN}$ that process each person are parametrized by the same weights.

To enable dyadic interaction, we facilitate information flow between motion signals by concatenating and processing the intermediate representations:

$$\hat{\mathbf{x}}_a^{(t)} = m_{\text{cross}}\big(\text{AdaLN}(d(\bar{\mathbf{x}}_a^{(t)} \oplus \bar{\mathbf{x}}_b^{(t)}) | \mathbf{c}, t)\big) \tag{5}$$

$$\hat{\mathbf{x}}_b^{(t)} = m_{\text{cross}}\big(\text{AdaLN}(d(\bar{\mathbf{x}}_b^{(t)} \oplus \bar{\mathbf{x}}_a^{(t)}) | \mathbf{c}, t)\big), \tag{6}$$

where $\oplus$ denotes dimension-wise concatenation and $d$ represents a learnable down-projection that ensures dimensional compatibility after concatenation. As in the per-person case, the functions $m_{\text{cross}}$, $\text{AdaLN}$, and $d$ that process each person are parametrized by the same weights.

Importantly, the non-commutative nature of concatenation enables $m_{\text{cross}}^{n}$ to differentiate between self-motion and partner motion, facilitating more effective interaction modeling. Each Mamba module consists of a residual stack of two Mamba models.

**Conditioning:** In each block, we condition the model on the diffusion time step $t$ and text embedding $\mathbf{c}$ through Adaptive LayerNorm (Adaptive LN) [22, 24]. We first project both conditioning signals and combine them additively to obtain a unified conditioning embedding $\text{emb} \in \mathbb{R}^h$, where $h$ represents the latent data dimension. For effective conditioning, we learn a projection of $\text{emb}$ that generates scale and shift parameters, which modulate the input signal. This signal modulation occurs prior to each Mamba module in the processing pipeline. The detailed conditioning mechanism is illustrated in Figure 5 (b).

### 3.2. Implementation Details

We implement our Dyadic Mamba with 8 blocks and latent dimension $h = 512$, which is less than half the number of parameters used in InterGen [18]. For each Mamba module, we use an expansion factor of 2 and a convolutional kernel size of 4. Similar to InterGen [18], we utilize a frozen CLIP-ViT-L/14 model to obtain text encoding $c \in \mathbb{R}^{768}$, which is randomly masked by 10% during training. For diffusion model training, we adopt a cosine noise schedule [20]. During training, we set the number of diffusion steps to $T = 1000$, while during inference, we sample via DDIM [33] using 50 steps. We evaluate our model on two datasets: InterHuman [18] and Inter-X [44].

InterHuman represents the data using a custom over-determined representation, combining SMPL [19] parameters, 3D joint coordinates, and foot contact binary variables, while Inter-X utilizes SMPL-X [21] with pose and global transformation. For both datasets, we ensure that $\mathbf{x}_a$ starts at the origin, facing the $x$-axis, and transform $\mathbf{x}_b$ accordingly. For training on InterHuman, we use their loss functions optimized for the over-parameterized representation.

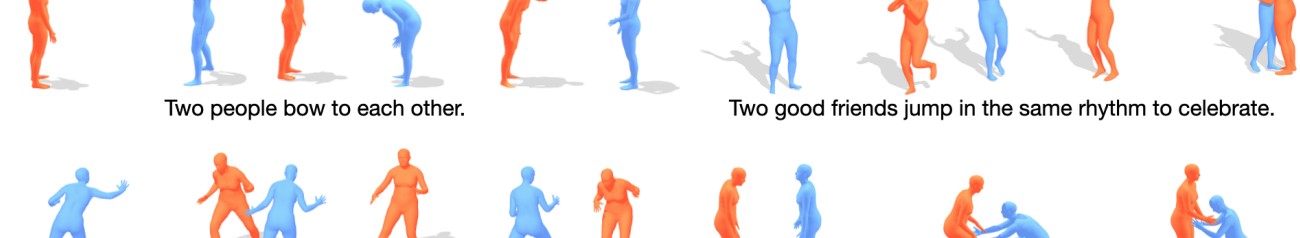

Two people bow to each other.

Two good friends jump in the same rhythm to celebrate.

Two fencers engage in a thrilling duel, their sabres clashing and sparking as they strive for victory.

One person is kneeling down while the other holds their hands to comfort them.

Figure 3. **Qualitative Results:** Dyadic motion generation results for various text descriptions.

| Dataset | Method | R Precision↑ | | | FID ↓ | MM Dist.↓ | Diversity→ | MModality ↑ | Arbitrary length |
| | | Top 1 | Top 2 | Top 3 | | | | | |
| --- | --- | --- | --- | --- | --- | --- | --- | --- | --- |
| | Real | $0.452^{\pm.008}$ | $0.610^{\pm.009}$ | $0.701^{\pm.008}$ | $0.273^{\pm.007}$ | $3.755^{\pm.008}$ | $7.948^{\pm.064}$ | - | - |
| InterHuman [18] | TEMOS [25] | $0.224^{\pm.010}$ | $0.316^{\pm.013}$ | $0.450^{\pm.018}$ | $17.375^{\pm.043}$ | $6.342^{\pm.015}$ | $6.939^{\pm.071}$ | $0.535^{\pm.014}$ | ✗ |
| | T2M [10] | $0.238^{\pm.012}$ | $0.325^{\pm.010}$ | $0.464^{\pm.014}$ | $13.769^{\pm.072}$ | $5.731^{\pm.013}$ | $7.046^{\pm.022}$ | $1.387^{\pm.076}$ | ✗ |
| | MDM [39] | $0.153^{\pm.012}$ | $0.260^{\pm.009}$ | $0.339^{\pm.012}$ | $9.167^{\pm.056}$ | $7.125^{\pm.018}$ | $7.602^{\pm.045}$ | $\mathbf{2.355^{\pm.080}}$ | ✗ |
| | ComMDM* [31] | $0.067^{\pm.013}$ | $0.125^{\pm.018}$ | $0.184^{\pm.015}$ | $38.643^{\pm.098}$ | $14.211^{\pm.013}$ | $3.520^{\pm.058}$ | $0.217^{\pm.018}$ | ✗ |
| | ComMDM [31] | $0.223^{\pm.009}$ | $0.334^{\pm.008}$ | $0.466^{\pm.010}$ | $7.069^{\pm.054}$ | $6.212^{\pm.021}$ | $7.244^{\pm.038}$ | $1.822^{\pm.052}$ | ✗ |
| | RIG [31] | $0.285^{\pm.010}$ | $0.409^{\pm.014}$ | $0.521^{\pm.013}$ | $6.775^{\pm.069}$ | $5.876^{\pm.018}$ | $7.311^{\pm.043}$ | $2.096^{\pm.065}$ | ✗ |
| | FreeMotion [6] | $0.326^{\pm.003}$ | $0.462^{\pm.006}$ | $0.544^{\pm.006}$ | $6.740^{\pm.130}$ | $3.848^{\pm.002}$ | $7.828^{\pm.130}$ | $1.226^{\pm.046}$ | ✗ |
| | Social Diffusion [37] | $0.341^{\pm.007}$ | $0.459^{\pm.011}$ | $0.544^{\pm.009}$ | $15.639^{\pm.090}$ | $3.856^{\pm.009}$ | $7.729^{\pm.020}$ | $1.030^{\pm.003}$ | ✓ |
| | InterGen [18] | $0.371^{\pm.010}$ | $0.515^{\pm.012}$ | $0.624^{\pm.010}$ | $5.918^{\pm.079}$ | $5.108^{\pm.014}$ | $7.387^{\pm.029}$ | $\underline{2.141^{\pm.063}}$ | ✗ |
| | InterGen [18] (RoPE) | $0.379^{\pm.005}$ | $0.531^{\pm.005}$ | $0.622^{\pm.005}$ | $6.002^{\pm.110}$ | $4.821^{\pm.010}$ | $7.776^{\pm.208}$ | $2.009^{\pm.014}$ | (✗)* |
| | InterMask [13] | $\underline{0.449^{\pm.004}}$ | $\underline{0.599^{\pm.005}}$ | $\underline{0.683^{\pm.004}}$ | $\mathbf{5.154^{\pm.061}}$ | $\mathbf{3.790^{\pm.002}}$ | $\mathbf{7.944^{\pm.033}}$ | $1.737^{\pm.020}$ | ✗ |
| | Ours | $\mathbf{0.458^{\pm.015}}$ | $\mathbf{0.607^{\pm.015}}$ | $\mathbf{0.685^{\pm.012}}$ | $\underline{5.792^{\pm.171}}$ | $3.793^{\pm.003}$ | $\underline{7.911^{\pm.048}}$ | $0.845^{\pm.055}$ | ✓ |
| | Real | $0.429^{\pm.004}$ | $0.626^{\pm.003}$ | $0.736^{\pm.003}$ | $0.002^{\pm.002}$ | $3.536^{\pm.013}$ | $9.734^{\pm.078}$ | - | - |
| Inter-X [44] | TEMOS [25] | $0.092^{\pm.003}$ | $0.171^{\pm.003}$ | $0.238^{\pm.002}$ | $29.258^{\pm.064}$ | $6.867^{\pm.013}$ | $4.738^{\pm.078}$ | $0.672^{\pm.041}$ | ✗ |
| | T2M [10] | $0.184^{\pm.010}$ | $0.298^{\pm.006}$ | $0.396^{\pm.005}$ | $5.481^{\pm.382}$ | $9.576^{\pm.006}$ | $5.771^{\pm.151}$ | $2.761^{\pm.042}$ | ✗ |
| | MDM [40] | $0.203^{\pm.009}$ | $0.329^{\pm.007}$ | $0.426^{\pm.005}$ | $23.701^{\pm.057}$ | $9.548^{\pm.007}$ | $5.856^{\pm.077}$ | $\underline{3.490^{\pm.061}}$ | ✗ |
| | MDM (GRU) [40] | $0.179^{\pm.006}$ | $0.299^{\pm.005}$ | $0.387^{\pm.007}$ | $32.671^{\pm.122}$ | $9.557^{\pm.019}$ | $7.003^{\pm.134}$ | $3.430^{\pm.035}$ | ✗ |
| | ComMDM [31] | $0.090^{\pm.002}$ | $0.165^{\pm.004}$ | $0.236^{\pm.004}$ | $29.266^{\pm.067}$ | $6.870^{\pm.017}$ | $4.734^{\pm.067}$ | $0.771^{\pm.053}$ | ✗ |
| | InterGen [18] | $0.207^{\pm.004}$ | $0.335^{\pm.005}$ | $0.429^{\pm.005}$ | $5.207^{\pm.216}$ | $9.580^{\pm.011}$ | $7.788^{\pm.208}$ | $\mathbf{3.686^{\pm.052}}$ | ✗ |
| | InterMask [13] | $\mathbf{0.403^{\pm.005}}$ | $\mathbf{0.595^{\pm.004}}$ | $\mathbf{0.705^{\pm.005}}$ | $\mathbf{0.399^{\pm.013}}$ | $\mathbf{3.705^{\pm.017}}$ | $\mathbf{9.046^{\pm.073}}$ | $2.261^{\pm.081}$ | ✗ |
| | Ours | $\underline{3.658^{\pm.007}}$ | $\underline{0.463^{\pm.007}}$ | $\underline{0.665^{\pm.008}}$ | $\underline{4.108^{\pm.018}}$ | $\underline{4.103^{\pm.019}}$ | $\underline{8.782^{\pm.053}}$ | $2.721^{\pm.070}$ | ✓ |

Table 1. Comparison of different methods on the InterHuman [18] and Inter-X [44] test set. **Bold** denotes the best results, while underline indicates the second-best results. * Note that InterGen with RoPE can generate motion sequences longer than seen during training: however, it cannot extrapolate to arbitrary sequence length.

For Inter-X, we employ a simple diffusion reconstruction loss on the SMPL-X parameters, combined with a forward-kinematics loss.

# 4. Experiments

## 4.1. Datasets

We evaluate our proposed Dyadic Mamba on two Text-to-Dyadic Motion benchmarks:
**InterHuman**: InterHuman [18] is a large-scale text-to-dyadic human motion dataset with $7,779$ motion sequences, with a total length of about $6.56$ hours. SMPL [19] body

meshes are fitted to both persons in each sequence and three text descriptions are given per sequence. The dataset is recorded at 60Hz, but baseline methods and evaluation protocol operate on 30Hz. We follow the evaluation protocol introduced in the original work.

**Inter-X**: Inter-X [44] is a large-scale text-to-dyadic human motion dataset with $11,388$ motion sequences, with a total length of $18.8$ hours. SMPL-X [19, 21] meshes are fitted to both persons, including hand and finger motion. Three text descriptions are given per sequence. We follow the evaluation protocol introduced in the original work.

## 4.2. Qualitative Results

In Figure 1, we present qualitative results of our method and compare them to the state-of-the-art model Inter-Gen [18]. Both models generate diverse and realistic motion for sequences up to 10s, which aligns with the training duration. However, beyond 10s, InterGen begins to exhibit motion artifacts, such as flickering and collapsing into unrealistic poses over extended time horizons. This limitation arises from the positional encoding, which constrains the model to the observed time horizon. In contrast, our method continues to produce realistic motion well beyond the sequence lengths observed during training. Additionally, our approach maintains close and realistic contact between both actors, even over long durations (Figure 1, first row). We provide additional qualitative samples generated from our method in Figure 3.

## 4.3. Quantitative Evaluation

We follow the evaluation protocols established for InterHuman [18] and Inter-X [44], which adapt the text-to-single-person motion metrics [10] to dyadic setups. Both methods provide pretrained feature extractors to obtain latent text and dyadic motion embeddings. We utilize the following evaluation metrics:

**R-Precision** is employed to evaluate the consistency between text and motion. The Euclidean distances between the motion and text embeddings are ranked, and the Top-1, Top-2, and Top-3 accuracies of motion-to-text retrieval are reported.

**Frechet Inception Distance** (FID) [11] is utilized to assess the similarity between synthesized and real motion distributions by calculating the distance between the latent embedding distributions of the generated and real interactive motions.

**The Multimodal Distance** (MM Dist) measures the distance between each text and its corresponding motion in latent space.

**Diversity** is calculated as the average Euclidean distance of 300 random samples of motion embeddings.

**Multimodality** (MModality) is calculated by randomly generating 20 samples per text prompt, randomly pairing them up, and reporting the average latent Euclidean distance between the pairs.

We report our results on short-term motion synthesis on InterHuman [18] and on Inter-X [44] in Table 1, where our method produces competitive results and outperforms other data-space based method.

## 4.4. Long-term Motion Synthesis

Current motion evaluation methods rely on pre-trained feature extractors that compress the entire motion sequence into a single vector. While this is highly useful for in-domain evaluation, the embedding space [18, 44] is trained

| Methods | $7s$ | $14s$ | $28s$ |
|---|---|---|---|
| Real | | 0.451±0.152 | |
| InterGen [18] | 0.343±0.170 | 0.290±0.118 | 0.271±0.088 |
| InterGen [18] (RoPE) | 0.322±0.020 | 0.326±0.016 | 0.212±0.009 |
| Ours | **0.365±0.159** | **0.379±0.157** | **0.376±0.155** |

Table 2. Average long-term per-person motion quality (NDMS [36] ↑) on InterHuman [18].

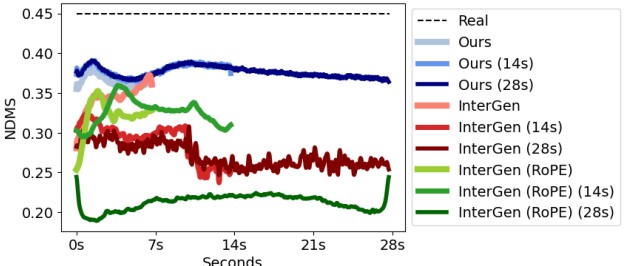

Figure 4. Per-frame long-term per-person motion quality (NDMS [36] ↑) evaluated on the InterHuman dataset [18] across multiple temporal horizons (7s, 14s, and 28s). Our approach demonstrates consistent performance compared to InterGen [18] and its variant InterGen (RoPE) [18, 34]. While InterGen exhibits degradation when generating sequences beyond its 10s training horizon, manifesting as increasingly apparent motion artifacts, our method maintains realistic motion quality across all evaluated temporal scales. The RoPE-enhanced variant successfully extends InterGen's effective generation window to approximately twice the training sequence length (≈20s), but fails to maintain coherence when generating longer sequences (28s). All three models were trained on sequence length of at most 10s.

on sequences with at most $10s$ length, meaning that long-term motion sequences are out-of-distribution. Furthermore, transformer-based approaches, such as InterGen [18], often produce motion artifacts when generating sequences with long time horizons beyond the maximum length seen during training (see Figure 4). These temporal effects are difficult to capture with a single compressed representation but are crucial for understanding a model's long-term synthesis capability.

To effectively capture and quantify these artifacts, we employ per-person per-frame Normalized Directional Motion Similarity (NDMS) [36]. NDMS is computed over a motion window size of $\frac{1}{3}$ seconds, as recommended by the authors. A higher NDMS score indicates that the structure and motion of the generated sample closely resemble those of the test set, thereby serving as a robust metric for evaluating individual motion quality.

To address the sequence length limitations of transformer-based models, we extend InterGen by replacing the Absolute Positional Embeddings with Rotary Positional Embeddings (RoPE) [34], which theoretically enables extrapolation beyond the sequence lengths ob-

| #Param | | | Conditioning | | Cross-Conditioning | | R Precision↑ | | | FID ↓ | MM Dist.↓ | Diversity→ | MModality ↑ |
|---|---|---|---|---|---|---|---|---|---|---|---|---|---|
| S | M | L | AdaLN | Prepending | + | ⊕ | Top 1 | Top 2 | Top 3 | | | | |
| | | | | Real | | | $0.452^{\pm.008}$ | $0.610^{\pm.009}$ | $0.701^{\pm.008}$ | $0.273^{\pm.007}$ | $3.755^{\pm.008}$ | $7.948^{\pm.064}$ | - |
| | ✓ | | ✓ | | ✓ | | $0.430^{\pm.005}$ | $\underline{0.589^{\pm.006}}$ | $\underline{0.667^{\pm.003}}$ | $6.531^{\pm.092}$ | $\underline{3.789^{\pm.002}}$ | $7.910^{\pm.032}$ | $\underline{0.948^{\pm.021}}$ |
| | ✓ | | | ✓ | | ✓ | $0.283^{\pm.005}$ | $0.417^{\pm.005}$ | $0.494^{\pm.004}$ | $7.301^{\pm.091}$ | $\mathbf{3.787^{\pm.003}}$ | $8.060^{\pm.025}$ | $\mathbf{1.322^{\pm.023}}$ |
| | ✓ | | ✓ | ✓ | | ✓ | $0.425^{\pm.004}$ | $0.567^{\pm.006}$ | $0.638^{\pm.004}$ | $6.817^{\pm.093}$ | $3.802^{\pm.002}$ | $\mathbf{7.930^{\pm.033}}$ | $0.909^{\pm.034}$ |
| ✓ | | | ✓ | | | ✓ | $0.427^{\pm.012}$ | $0.576^{\pm.012}$ | $0.654^{\pm.010}$ | $\mathbf{5.577^{\pm.172}}$ | $3.804^{\pm.002}$ | $7.855^{\pm.045}$ | $0.915^{\pm.049}$ |
| | ✓ | | ✓ | | | ✓ | $\underline{0.432^{\pm.006}}$ | $0.574^{\pm.004}$ | $0.649^{\pm.005}$ | $5.821^{\pm.088}$ | $3.802^{\pm.001}$ | $7.879^{\pm.029}$ | $0.861^{\pm.025}$ |
| | | ✓ | ✓ | | | ✓ | $\mathbf{0.458^{\pm.015}}$ | $\mathbf{0.607^{\pm.015}}$ | $\mathbf{0.685^{\pm.012}}$ | $\underline{5.792^{\pm.171}}$ | $3.793^{\pm.003}$ | $\underline{7.911^{\pm.048}}$ | $0.845^{\pm.055}$ |

Table 3. Ablation of our key designs on InterHuman [18] test set. **Bold** denotes the best results, while underline indicates the second-best results.

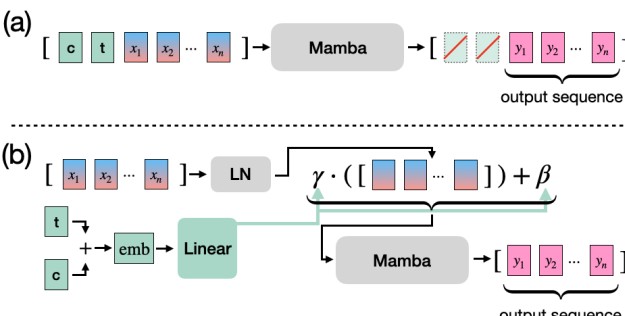

Figure 5. **Conditioning:** We experiment with two variants for conditioning the Dyadic Mamba on diffusion step $t$ and text embedding $\mathbf{c}$: (a) Prepending approach: The conditioning embeddings are simply prepended to the motion sequence before passing it to the Mamba module. (b) Adaptive LayerNorm modulation: The input signal is scaled and shifted by a linearly projected addition of $t$ and $\mathbf{c}$ before being passed to the Mamba module.

served during training. We re-train this modified model (InterGen (RoPE)) on the InterHuman dataset and report the results on standard benchmarks in Table 1.

In Figure 4, we compare the per-frame motion generation quality of our method and InterGen for three time horizons: 7s, 14s, and 28s. Both InterGen and our method are trained on sequences with a maximum length of 10s. Our method successfully extrapolates beyond the sequence length observed during training, while InterGen produces artifacts for longer sequences (see Figure 1 last row at 28s). Moreover, we observe that for InterGen, longer motion sequences degrade the quality of motion even within the initial 10s window. This degradation is attributed to the Transformer encountering positional encodings that were not observed during training.

In Table 2, we report the average NDMS scores over the three time horizons. When utilizing RoPE, InterGen demonstrates improved capability to generate sequences beyond the training length. However, when generating sequences longer than $2\times$ the sequence length seen during training, the model's performance deteriorates significantly, which aligns with the limitations of RoPE documented in the language modeling domain [15, 23, 28].

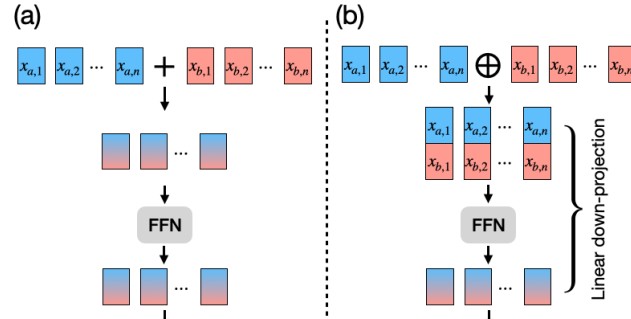

Figure 6. **Cross-Conditioning:** We experiment with two variants for cross-person information flow: (a) Simple addition: The intermediate motion representations from both individuals are combined through element-wise addition. This approach is order-invariant. This property could potentially facilitate extension to multi-person interactions beyond dyadic scenarios. (b) Concatenation and down-projection: The intermediate motion representations are concatenated along the feature dimension and then projected back to the original dimensionality through a learned linear transformation. This approach preserves the identity of each person in the interaction, allowing the model to distinguish between self-motion and partner motion, but introduces order-dependency in the representation.

Note that our approach only evaluates individual motion quality and per-frame human interaction evaluation remains an open problem for future work.[1]

## 4.5. Ablation Study

In this section, we analyze the impact of key design choices in our approach. All experiments are conducted on the InterHuman dataset [18], with results reported in Table 3.

**# Parameters**: To determine optimal parameter efficiency, we experiment with three model variants: **L** ($248M$ parameters, $\times 1.36$ InterGen), **M** ($79M$ parameters, $\times 0.43$ InterGen) and **S** ($32M$ parameters, $\times 0.18$ InterGen). **M** outperforms **S** on R-Precision while **S** achieves better FID scores.

---

[1]We encourage reviewers to check our supplementary video, where we show that our model can generate realistic social interactions beyond one minute.

We select the **M** variant for our final model, prioritizing text-to-motion fidelity as the most critical factor for the text-to-dyadic motion synthesis task.

**Impact of Conditioning**: Conditioning on both the diffusion step $t$ and the conditioning vector $\mathbf{c}$ in diffusion models remains an open problem, with various techniques proposed. In transformer-based architectures, cross-attention is commonly employed. However, recent studies have demonstrated that Feature-wise Linear Modulation (FiLM) [24] and its adaptations such as Adaptive LayerNorm [22] outperform cross-attention methods not only in image generation but also in dyadic motion synthesis [18].

To identify the optimal conditioning method for Mamba-based models, we conduct a systematic comparison of three conditioning variants: (1) Prepending the conditioning vector $\mathbf{c}$ and diffusion step $t$ to the time series $\mathbf{x}$ (see Figure 5 (a)), (2) Using Adaptive LayerNorm modulation (see Figure 5 (b)), (3) and the combination of (1) and (2).

As shown in Table 3, Adaptive LayerNorm (AdaLN) significantly outperforms both the simple prepending approach and the combined strategy. This finding aligns with similar observations in the image generation domain [22], suggesting that the benefits of AdaLN's modulation approach extend effectively to state-space models for motion synthesis tasks.

**Impact of Cross-Conditioning**: Replacing cross-attention in Mamba is an open problem, and in this section we analyze two simple yet effective approaches: (1) addition of the two intermediate motion sequences $\overline{\mathbf{x}}_a + \overline{\mathbf{x}}_b$, as shown in Figure 6 (a) or (2) concatenation and down-projection, as shown in Figure 6 (b). We insert an additional linear projection to variant (1) to ensure equal parameter count between the two versions. We conjecture that these simple per-frame operations are effective because both signals are temporally aligned, and we note that this approach might not work for signals without proper temporal alignment, such as texts or images.

An important distinction between these methods is that addition is an order-invariant operation while concatenation is not. Interestingly, addition performs very competitively when compared to concatenation, as shown in Table 3, especially on R-Precision metrics. This suggests that addition is a viable strategy for information passing and might even enable more-than-dyadic social interaction modeling, leveraging its order-invariant property. However, for our final model architecture, we selected concatenation as it yielded the best FID score, indicating superior overall motion quality and fidelity to the distribution of real dyadic interactions.

### 4.6. Failure Cases

In Figure 7 we present some typical failure cases to highlight potential for improvement. For example, persons might sometimes clip through each other or change order, as

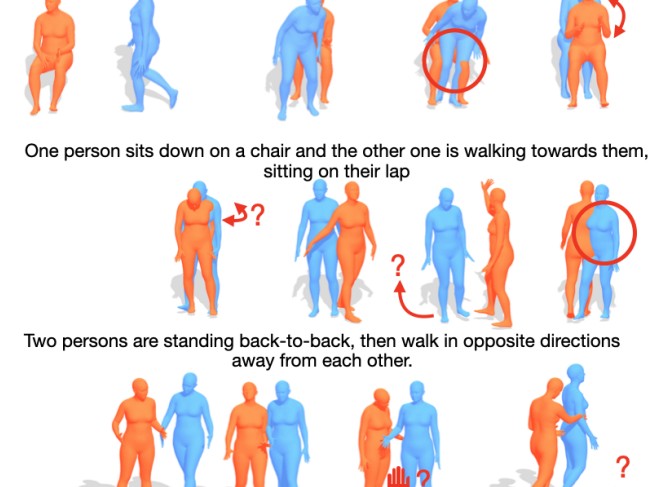

One person sits down on a chair and the other one is walking towards them, sitting on their lap

Two persons are standing back-to-back, then walk in opposite directions away from each other.

The two persons are walking hands in hand in a straight line.

Figure 7. **Failure Cases**: First row: if one person is static (i.e. sitting) and the other person is instructed to interact with them, they might clip through each other and/or change order. Second + third row: out-of-distribution text: the persons are not back-to-back but face the same directions, only one person walks, and they are walking towards the other person, not away. The persons turn even though they are instructed to walk in a *straight* line.

can be seen in the first sitting example in the first row and in the turning example in the second row. The model also has a limited understanding for out-of-distribution descriptions, i.e. it cannot generate persons *back-to-back* and it cannot generate motion that goes on in a single direction and instead reverses towards the center, adhering to the training data distribution. Note that these kinds of artifacts are expected and can also be observed in other transformer-based methods.

## 5. Conclusion

In conclusion, we have introduced Dyadic Mamba, a novel approach for synthesizing dyadic human motion from text. By leveraging a simple yet effective stacked Mamba-based architecture, our method produces competitive results on two dyadic motion datasets and addresses the limitations of existing transformer-based models, particularly in handling long sequences. The ability to generate realistic and coherent motion over extended time horizons, while maintaining parameter efficiency, highlights the effectiveness of our approach for modeling complex social interactions.

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
