# OpenReview forum: "Dyadic Mamba: Long-term Dyadic Human Motion Synthesis"
_thecvf.com/CVPR/2025/Workshop/HuMoGen — CVPR 2025 Workshop HuMoGen Submission_

### Official Review · Reviewer_ziZx · 2025-03-27
**Dyadic Mamba: Long-term Dyadic Human Motion Synthesis**

**Rating:** 3
**Confidence:** 4

**Review:**

The paper proposes Dyadic Mamba, which introduces State-Space Models (SSMs) into the motion synthesis domain to generate high-quality dyadic human motion of arbitrary length. In addition, it introduces a new metric, Normalized Directional Motion Similarity (NDMS), to evaluate long-term motion quality.

Strength：
1. The combination of the self-mamba and cross-mamba modules for modeling dyadic interaction is novel and effective.
2. Incorporating RoPE into InterGen for further comparison is meaningful and provides additional evidence of the effectiveness of this work.
3. Both quantitative and qualitative experiments demonstrate the effectiveness of the proposed approach.

Weakness：
1. In Table 1, the performance improvement of Dyadic Mamba over prior methods appears limited in some metrics. For instance, in the Inter-X dataset, the Top-1 accuracy of the proposed method is significantly lower—by an order of magnitude—compared to other methods. Could the authors provide a more detailed analysis and explanation of these results?
2. As the proposed method shows comparable performance to InterMask on most metrics, including more qualitative comparisons with InterMask may better highlight the advantages of Dyadic Mamba.
3. Although the introduction of Mamba reduces the number of parameters, the paper does not discuss its inference speed. How does the proposed method perform in terms of generation speed?
4. There are minor formatting inconsistencies. For example, in Table 2, the term linearly projection d is italicized in most cases, but not consistently across the figures. Additionally, the word cross is unnecessarily bolded in Line 395.

---

### Official Review · Reviewer_gyVa · 2025-03-27

**Rating:** 3
**Confidence:** 4

**Review:**

**Summary:**

This paper addresses the problem of generating arbitrary-length two-person motion from text. The authors propose integrating Mamba modules into a diffusion-based text-to-two-person motion generation setting to overcome limitations in long-sequence generation for prior transformer-based models caused by positional encoding constraints. The proposed framework, DyadicMamba, achieves competitive two-person motion generation results on standard-length sequences and outperforms prior work on a newly proposed benchmark for long-term motion synthesis.


**Strengths:**
- The paper is organized and easy to follow. The motivation for using Mamba modules to achieve long-sequence generation is clear.
- The qualitative motion synthesis results show compelling improvement of motion quality and naturalness for long sequences compared to prior work.
- The proposed evaluation method for long sequence generation is reasonable.

**Weaknesses:**
- Mamba modules have been previously applied to human motion generation, including long-sequence single-person settings (e.g., Motion Mamba [ECCV 2024]) and standard two-person settings (e.g., TIMotion [CVPR 2025]). While the proposed method effectively uses this well-established component, it does not introduce new insights beyond its application. There may be potential to further refine the Mamba module for long sequence two-person motion synthesis, but the paper primarily adopts a default setup without enhancing the module itself.


- The authors present a reasonable baseline mode where they retrain transformer-based denoisers with RoPE to extrapolate to longer sequence lengths for motion generation. However, it should be noted that diffusion-based models (e.g., InterGen) and masked generative models (e.g., InterMask) already exhibit strong motion in-filling capabilities. An alternative inference-only baseline could employ a sliding window approach, where the end frame(s) or discrete state(s) of the previous window serves as the initial key frame(s) for generating the next window under the same text condition. While this approach may be less efficient, it would be interesting to see a comparison of motion quality in such a setting.


- The authors provide a nice discussion on the role of order-variant vs. order-invariant cross-conditioning in their ablation study. However, the ablation table lacks the M-size model with the AdaLN Norm + Addition setting. While the authors note an increase in FID at the cost of R-Precision when using concatenation in the L-size model, it would be beneficial to verify whether this trend holds in the M-size setting before concluding that Concatenation is the preferred choice over Addition for cross-conditioning.

**Overall:**
The paper presents good results but offers limited new insights. I lean toward a borderline rating.

---

### Meta-Review · Area_Chair_sz6W · 2025-03-29

**Recommendation:** Accept

**Metareview:**

Both reviewers rate this as a borderline paper (score: 3/5) with high confidence (4/5). They acknowledge the paper's technical soundness and the quality of results but question its novelty and contribution beyond applying Mamba to a new domain. The consistent evaluation from two independent reviewers strengthens the reliability of these assessments.
The primary achievement of this work lies in effectively adapting SSMs for dyadic motion generation with arbitrary length capability - addressing a practical limitation in existing approaches. While not revolutionary in its technical components, the integration demonstrates meaningful results for an important problem in motion synthesis.
The inconsistencies in performance metrics warrant further explanation. For instance, the significant drop in Top-1 accuracy on the Inter-X dataset needs clarification, as noted by reviewer ziZx. Additional qualitative comparisons with strong baselines like InterMask would strengthen the paper's case.

*Recommendation*:
Based on the reviews, I recommend acceptance with revisions. The paper presents valuable contributions to long-sequence dyadic motion generation. However, the authors are encouraged to provide clearer positioning of their contribution to existing Mamba applications in motion synthesis, and a more detailed analysis of performance variation.

---

### Decision · Program_Chairs · 2025-03-31

Accept